# Ecological Zoning of the Baikal Basin Based on the Results of Chemical Analysis of the Composition of Atmospheric Precipitation Accumulated in the Snow Cover

Yelena V. Molozhnikova *, Maxim Yu. Shikhovtsev, Olga G. Netsvetaeva and Tamara V. Khodzher *

Limnological Institute, Siberian Branch, Russian Academy of Sciences, Ulan-Batorskaya Street 3, Irkutsk 664033, Russia; max97irk@yandex.ru (M.Y.S.)
* Correspondence: yelena@lin.irk.ru (Y.V.M.); khodzher@lin.irk.ru (T.V.K.)

**Abstract:** This research used the geostatistical analysis of snow cover samples taken in 2017–2022 in the Baikal basin. Groups of snow cover pollution sources were identified by the method of empirical Bayesian kriging (ArcMap software) and mathematical data processing. The studied area was divided into fourteen districts. Geovisualization of marker substances accumulated in the snow cover allowed for the zoning of the studied area according to the degree of anthropogenic load. It was revealed that the atmospheric pollution of the territory from local sources extended for tens of kilometers along the prevailing wind direction. The maximum concentrations of anthropogenic aerosols in the snow cover were determined in towns that were sources of pollution and near settlements located on the coast of Lake Baikal and at the Selenga River mouth. The industrial centers of the region and the southern basin of Baikal, being affected by the air emissions from the Irkutsk agglomeration, were determined to be the most susceptible to anthropogenic pollution. The middle and northern basins could be attributed to the background regions being affected only by local heating sources and the natural background. The main atmospheric pollutants and the areas of their distribution were established. The main sources of snow cover pollution in the region, in addition to the natural background, were emissions from thermal power plants and motor vehicles.

**Keywords:** air pollutants; snow cover; the Baikal natural territory; geostatistical analysis





## 1. Introduction

Studying the chemical composition of the snow cover is used for efficient ecological and economic monitoring of air pollution in regions with a cold climate [1]. The snow cover has long been investigated to estimate the transfer of atmospheric pollutants from natural and anthropogenic sources [2–4]. The studies showing the processes of long-term pollution of the underlying surface both in industrial regions and in background areas remote from them are of particular interest. Snow cover is a complex heterogeneous system, the main components of which are solid water (whose content usually exceeds 99%) and an admixture of solid aerosol particles [4]. These aerosol fractions are of the greatest interest in the study of snow since their characteristics can be used to obtain information about the sources and chemical composition of aerosols and the nature of the distribution of anthropogenic impurities emitted into the atmosphere and transferred over long distances [5]. There is no general classification of the sources of aerosol particles in snow. It can be assumed that some aerosol particles are of natural origin (continental and marine aerosol, cosmic dust) [6,7], and others come from anthropogenic emissions from industrial enterprises, transport, agriculture, and utilities [8–11]. Depending on the sampling sites, the contribution of pollution sources and the ratios between substances can differ in chemical composition and the predominance of marker substances [12,13].

Currently, snow cover is increasingly used in Russia as an object for monitoring the state of the atmosphere as well as an integral indicator of atmospheric pollution in areas

characterized by the presence of stable snow cover for a long time. Studies on snow cover monitoring can be divided into several groups: local, regional, or global. Most of them are devoted to the state of the environment in industrial centers [14–20]. Some studies assess the impact of a specific industrial group of sources [21–26] or a certain anthropogenic substance [27,28] on the environment by means of snow cover analysis. Much fewer studies are devoted to regional monitoring in large areas [29–36] and in background regions [37,38], which is associated with the laboriousness of selecting specific samples over large areas.

Lake Baikal is the world's largest reservoir of pure fresh water (contains 23,000 km$^3$). The significance of the lake as a world natural heritage site is defined in UNESCO documents and the Federal Law of 1 May 1999 no. 94-FZ "On the Protection of Lake Baikal". Due to the ongoing development of industry in the region, the expansion of the infrastructure of tourist and recreational areas of the coastal zone of Lake Baikal, anthropogenic impacts on various ecosystems of the Baikal natural territory and the lake have increased in recent years. The state of atmospheric air over Baikal is determined not only by its geographical location and climatic features but also by anthropogenic emissions from objects located along the coast and the main air mass transfer pathways from the industrial centers of the Baikal region. An analysis of the studies of the chemical composition of the snow cover in the Baikal region has shown their rather large fragmentation: the studies were performed mainly in separate areas, not covering large areas of the Baikal basin. In this study, for the first time, the assessment of snow cover pollution in the vast territories of the Baikal region, including industrial centers (cities), rural areas, and background areas, was carried out.

This research was aimed at performing ecological zoning in the Baikal basin using the results of the chemical composition of atmospheric precipitation accumulated in the snow cover. In order to achieve this aim, a detailed analysis of the chemical composition of the snow cover in the large industrial centers of the Baikal region, affecting the overall level of pollution in the region and rural and background areas, was performed using the method of geostatistical data processing. When performing the research, the published data of the employees of the Limnological Institute [39–46], as well as other research organizations [47–52], and the results of the authors' own studies over the past six years were used.

## 2. Sampling and Analysis Methods

The research area was located in the center of Asia (103°39′–110°42′ E, 51°15′–55°55′ N). The research object was the central ecological zone of Lake Baikal (water area and coast), Barguzinskaya valley, as well as areas of industrial centers of the Baikal region affecting the lake. Sampling sites were selected based on the preliminary HYSPLIT model calculations while taking into account the sources of pollutants, prevailing winds, and the surrounding terrain. The pH value and electrical conductivity were determined in the samples of the snow cover. After filtration through a membrane filter, the concentrations of the main ions were measured. Determination of cations in atmospheric precipitation was carried out on the ContrAA800 atomic absorption spectrophotometer (Analytik Jena AG, Jena, Germany), anions—on the ICS-3000 ion system (Dionex, Sunnyvale, CA, USA). Chemical analyses were performed in the accredited laboratory of hydrochemistry and atmospheric chemistry of the Limnological Institute of the Siberian Branch of the Russian Academy of Sciences according to the methods recommended in the atmospheric monitoring networks of international programs—EMEP [53] and EANET [54]. In order to build maps, more than 400 receptor stations were analyzed, more than 180 of which were selected regularly in the monitoring mode over the past 4–6 years (Figure 1).

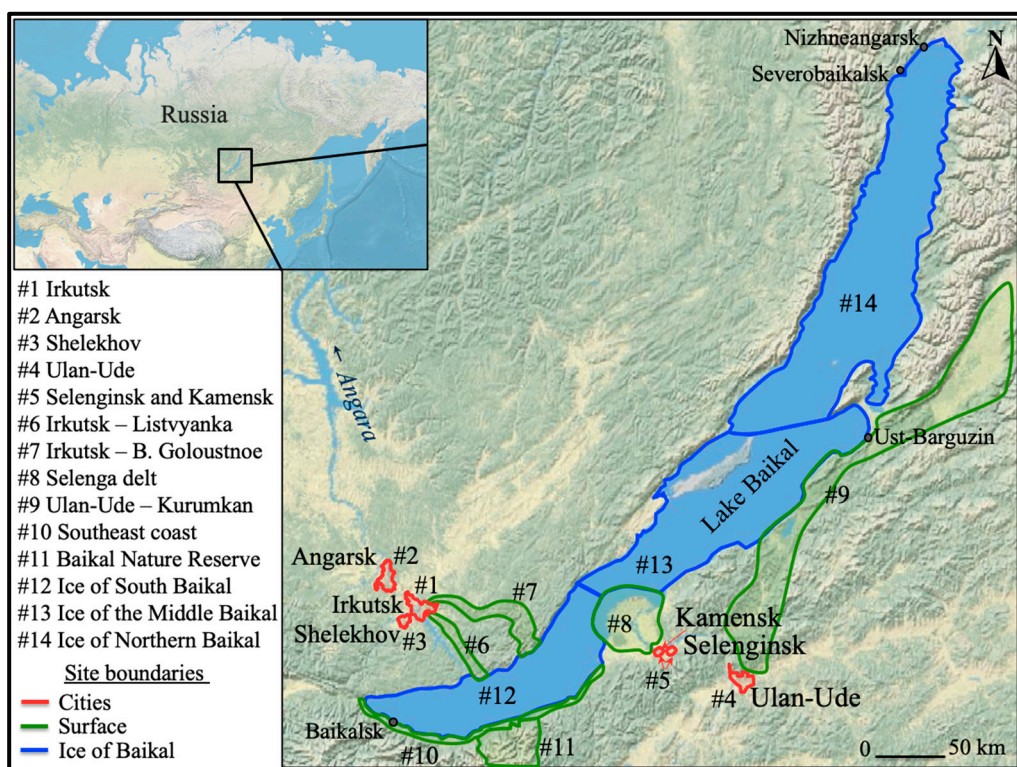

**Figure 1.** Location of sites where snow samples were taken in 2017–2022.

### 2.1. Field Sampling

Snow samples were collected on the territory of the Baikal region from 25 February to 10 March 2017–2022. Since the maximum depth of snow cover was defined at the end of February in the southern regions and early March in the northern regions, sampling was performed at the time when the total maximum accumulation of precipitation during the period of stable snow cover before the spring melt was defined. According to estimates of meteorological records for this region, this was from two to four months [55,56]. In total, over six years, more than 800 samples were taken by the researchers of the Limnological Institute. In order to cover a larger area, the research also used data obtained from open sources [47–52]. Snow samples were taken using a polycarbonate sampler. Particular attention was paid to the snow sampling near the ground surface in order to exclude the impact of substances from the soil and vegetation cover on the chemical composition of the snow. The bottom layer of 5 to 10 cm was rejected. In order to avoid contamination of the new snow sample by the previous sample, the sampler was thoroughly cleaned from the previous sample by washing it with snow at the place of subsequent sampling. On the background sites, sampling was performed at a considerable distance from highways.

### 2.2. Location of Districts

For the convenience of data analysis and the construction of tracer substance accumulation maps, the entire research area was divided into fourteen districts (Figure 1). The first five districts were industrial cities that were sources of pollution and were located outside the central ecological zone of Baikal. Districts 6, 7, 9, and 10 were located along the roads in the direction of the main pathways of transfer of atmospheric emissions from the large industrial centers. The eighth group included samples from the delta of the Selenga River. This coastal area was located on the pathway of transfer of atmospheric emissions from the cities of Selenginsk and Kamensk (Figure 2). The eleventh group included snow cover samples taken on the territory of the Baikal State Natural Biosphere Reserve. No local anthropogenic sources were found on its territory, and most of its territory was located along the main pathway of transfer of atmospheric emissions from anthropogenic sources

located in the Angara River valley [46]. The samples taken from the ice of Lake Baikal were included in separate groups (from the twelfth to the fourteenth) due to different periods of accumulation of snow cover on the ice of the lake (two and a half months) and on its coast (four months) [55,56].

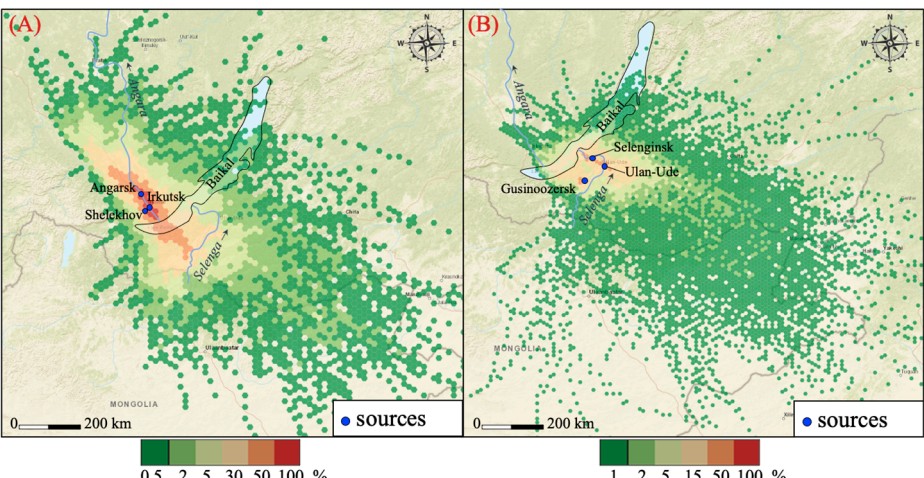

**Figure 2.** Percentage distribution of transfer pathways of air masses (250 m above ground level (AGL)), calculated from the main source cities of the Baikal region during 2017–2022 (**A**) Irkutsk Region (**B**) Republic of Buryatia https://www.ready.noaa.gov/HYSPLIT.php (accessed on 16 January 2023).

*2.3. Meteorological Parameters*

Wind direction and strength of the wind, as well as the temperature regime, are important factors affecting the distribution of emissions in the air [57,58]. Therefore, initially, when choosing reference stations, an analysis was made of the transfer of air masses from large stationary sources of atmospheric pollution (HYSPLIT model https://www.ready.noaa.gov/HYSPLIT.php (accessed on 16 January 2023)), and possible precipitation fields of atmospheric impurities (Figure 2) over the water area and coast of Lake Baikal were obtained. According to archival meteorological data (GDAS archive), for each day of the period (November–February), reverse trajectories of the movement of air masses were constructed. The duration of the trajectories was chosen to be 24 h. Calculations were performed for 250 m above ground level. Figure 2 demonstrates the distribution of air masses over Lake Baikal averaged over a six-year period: the main transfer pathway of air masses from stationary sources located in the Angara River valley (Irkutsk Region) was directed to the southeast coast of Lake Baikal (30–50%) and to the delta of the Selenga River (about 5%), which could contribute to a greater accumulation of anthropogenic impurities in the snow cover with a northwesterly wind direction. Transfer of impurities to the water area of Baikal from the industrial centers of the Republic of Buryatia occurred along the valley of the Selenga River in its delta (15%). For a more accurate visualization of the distribution of atmospheric impurities in the snow of the Southern Basin and the Selenga Delta, the sampling grid was made in more detail and was based on the calculations. The transfer of pollutants to the middle and northern basins of the lake, as calculations showed, was insignificant.

The identification of groups of sources affecting the chemical composition of atmospheric fallout was performed in two stages. The first stage included an inventory of all possible natural and anthropogenic sources of aerosols in the region (Table 1). In the second stage, the natural and anthropogenic groups of sources affecting the chemical composition of atmospheric fallouts were determined by the methods of statistical processing of data from atmospheric fallout samples [59,60], and the choice of tracer substances was made.

**Table 1.** Location and general characteristics of districts where snow cover samples were taken [61–66].

| Districts | Acreage, km² / Number of Receptor Points | Quantity of Atmospheric Precipitation, mm/Year, Snow Height, cm | Soils | Vegetation | Anthropogenic Sources |
|---|---|---|---|---|---|
| Irkutsk | $\frac{305}{37}$ | $\frac{485}{22}$ | Gray, dark gray, alluvial gray-humus, sod-podzolic | Larch-pine forests, young birch and aspen forests, shrub communities, meadow communities | Mechanical engineering and instrumentation, heat and power engineering, electric power industry, production of building materials, light and food industry, automobile, and railway transport |
| Angarsk | $\frac{294}{22}$ | $\frac{480}{20}$ | Sod-calcareous, sod-podzolic, gray forest, sod-gray | Pine forests, young birch and aspen forests, shrub communities, meadow communities | Chemical and petrochemical industry, thermal power engineering, production of building materials, light and food industry, automobile and railway transport |
| Shelekhov | $\frac{31}{24}$ | $\frac{485}{22}$ | Gray forest gleyic and gley, soddy-podzolic | Larch-pine forests, young birch and aspen forests, shrub communities, meadow communities | The main industries are non-ferrous metallurgy, thermal power engineering, motor transport |
| Ulan-Ude | $\frac{347}{28}$ | $\frac{280}{18}$ | Soddy-forest and sandy loamy soils, alluvial meadow marsh and meadow-marsh soils | Tree and shrub vegetation, meadow and steppe communities | The main industries are mechanical engineering and metalworking, thermal power engineering, automobile and railway transport |
| Selenginsk and Kamensk | $\frac{20}{20}$ | $\frac{440-460}{35}$ | Soddy forest gray | Tree and shrub vegetation, wet meadows, meadow and steppe communities | The main industries are woodworking, pulp and paper, construction, municipal energy, road transport |
| Irkutsk—Listvyanka | $\frac{260}{14}$ | $\frac{485-490}{30}$ | Gray, soddy-podzolic, podzolized podburs | Larch-pine forests, birch forests, shrub communities | Municipal boiler houses, house stoves, road transport |
| Irkutsk—Bolshoe Goloustnoye | $\frac{212}{15}$ | $\frac{485-240}{27}$ | Gray, soddy-podzolic, podzolized podburs | Larch-pine forests, steppes | Woodworking enterprises, communal boiler houses, house stoves, road transport |

Table 1. *Cont.*

| Districts | Acreage, km² / Number of Receptor Points | Quantity of Atmospheric Precipitation, mm/Year, Snow Height, cm | Soils | Vegetation | Anthropogenic Sources |
|---|---|---|---|---|---|
| Delta of the Selenga River | 540 / 26 | 440 / 11 | Alluvial gray-humus meadow bog, alluvial meadow carbonate, alluvial soddy | Grass swamps, wet meadows, pine forests | Municipal boiler houses, house stoves, road transport |
| Ulan-Ude Kurumkan | 5670 / 31 | 280–380 / 26 | Podburs, coarse humus burozems, alluvial gray humus, gray and light humus | Cedar-larch-pine forests, swamps | Municipal boilers, house stoves, road transport, agriculture |
| Southeastern shore | 796 / 24 | 640 / 50 | Podzolized podburs | Fir-spruce-cedar forest, birch forests | Production of building materials, municipal boiler houses, house stoves, road and railway transport |
| Baikal National Nature Reserve | 1657 / 8 | 1360 / 98 | Permafrost-taiga, soddy-lithogenic brown forest coarse humus burozems | Fir-spruce-cedar forests, alpine meadows, shrub communities | No anthropogenic sources are available |
| The coast of the southern basin of Baikal | 7200 / 49 | 430 / 12 | Podzolized podburs, coarse burozems, cryozems | Larch-pine forests, birch forests, swamp vegetation, fir-spruce-cedar forests | Municipal boiler houses, production of building materials, house stoves, road and railway transport |
| Coast of the middle basin of Baikal | 11,600 / 64 | 350 / 12 | Podburs, coarse humus, chestnut, soddy-podbrown burozems | Pine forests, larch forests, cedar-larch-pine forests, steppe vegetation, swamp vegetation | Municipal boiler houses, house stoves, road transport |
| The coast of the northern basin of Baikal | 12,700 / 53 | 330 / 11 | Podburs, podzols, cryozems, eutrophic peat soils, dark-humus carbon-lithozems | Cedar-larch-pine forests, dwarf pine, larch forests, birch forests, marsh vegetation | Municipal boiler houses, house stoves, railway and road transport |

*2.4. Geostatistical Analysis Method*

Geostatistics is considered a good tool for analyzing and interpreting spatially distributed data and a convenient method for assessing pollution and controlling environmental risks over a large area. Interpolation methods are widely used in mapping processes to estimate the accumulation of pollutants in areas where no sampling was performed. Due to various reasons (inaccessibility of places or the impossibility of sampling on the territory of industrial facilities), sampling could not be performed evenly. Therefore, empirical Bayesian kriging was chosen as a tool for mapping (ArcMap 10.2). It differs from other kriging methods since it takes into account the error associated with the estimation of the main semivariogram and provides accurate interpolation of moderately heterogeneous data, which allows visualizing data as close to the real picture as possible.

*2.5. Calculation of the Values of Accumulation in Snow Cover*

Quantitative assessment of the inflow of polluting marker substances into the studied area required calculation of the values of the accumulation of individual ions in the snow during the period of stable snow cover. The accumulation of marker substances was an integral characteristic of the contamination of the underlying surface, including the snow cover. It showed the amount of substances (mg, kg, t) entering per unit area of the underlying surface ($m^2$, $km^2$). The calculation of this value was performed according to the following equation:

$$Q = \alpha \cdot \times C \times W \tag{1}$$

where Q was the accumulation, $mg/m^2$; C was the concentration, $mg/L$; W was the moisture content, cm; $\alpha$ was the coefficient taking into account the dimension.

Moisture content is the amount of moisture determined at the sampling site. It was calculated by the following equation:

$$W = V/S \tag{2}$$

where V was the sample volume, $cm^2$; S was the sampling area, $cm^2$ [4].

**3. Results and Discussion**

*3.1. Concentration of the Main Ions in the Snow Waters of the Baikal Basin*

Based on the data on the concentrations of the main ions in snow water, boxplots were constructed. Figure 3 shows the results of the chemical analysis of meltwater samples taken in 2017–2022 at fourteen sites (Figure 1). In Figure 3, the ordinate shows the values of concentrations (in $mg/dm^3$), and the abscissa manifests the number of the region. The diagram shows the median (thick line), the first quartile (Q1, the lower limit of the "box"), and the third quartile (Q3, the upper limit of the "box"), as well as one-time outliers, which are atypical values that fall out of the general series of observations and exceed the limits of one and a half product differences between Q3 and Q1 (circles).

Despite the diversity of industrial facilities in the region (Table 1), their influence is local and concentration maxima are obtained near the source (Figure 3). As can be seen from Table 1, the main anthropogenic sources in the region are heat power facilities and motor transport [67]. Therefore, the concentrations of sulfates, the tracer of coal combustion, in years. Irkutsk, Angarsk, and Ulan-Ude have on average similar values: (median from 5.7 to 6.2 mg/L). Concentrations are similar to the results, as in the cities of the European part of Russia: 2.2–7.7 mg/L in Moscow [68] and 3.4–5.6 in Izhevsk [69] and the Far East: 4.9 mg/L, (Vladivostok) [70]. In areas remote from sources (plots No. 9, 10, 11, 13 14) median concentrations of sulfates vary from 0.01 to 0.95 mg/L. This is less or comparable to the concentrations of sulfates (0.86 mg/L) in the snow cover in the north of the country (Komi Republic) [71].

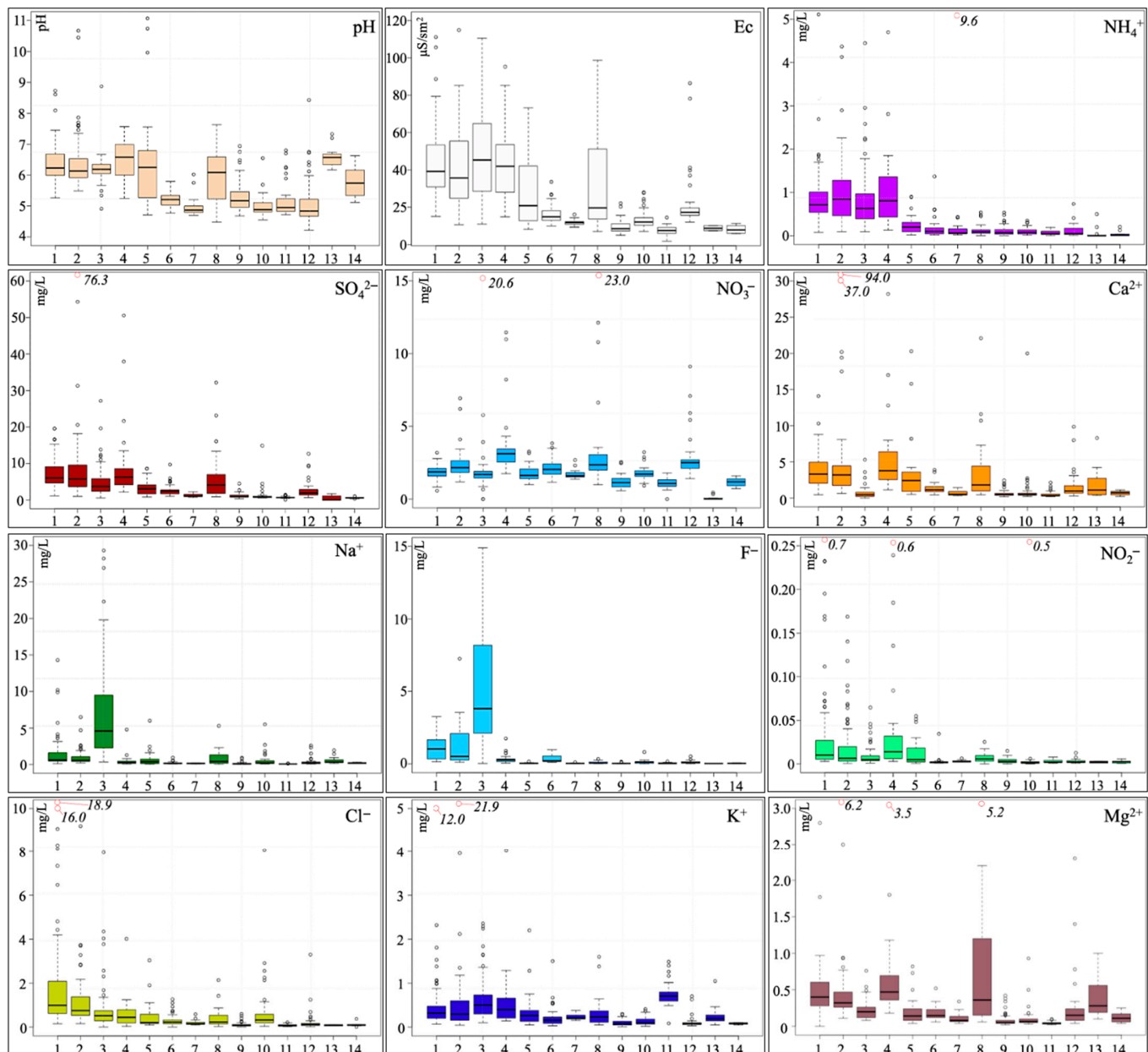

**Figure 3.** Concentrations of the main ions in the snow water of the Baikal Basin; Data analysis and discussion; Sampling area number: (1) Irkutsk; (2) Angarsk; (3) Shelekhov; (4) Ulan-Ude; (5) Selenginsk and Kamensk; (6) Irkutsk—Listvyanka; (7) Irkutsk—B. Goloustnoe; (8) Selenga delt; (9) Ulan-Ude—Kurumkan; (10) Southeast cost; (11) Baikal Nature Reserve; (12) Ice of South Baikal; (13) Ice of Middle Baikal; (14) Ice of Nothern Baikal. The diagram shows the median (thick line), the first quartile (Q1, the lower limit of the "box") and the third quartile (Q3, the upper limit of the "box"), as well as one-time outliers being atypical values that fall out of the general series of observations and exceeding the limits of one and a half product differences between Q3 and Q1 (circles).

Nitrogen oxides (Figure 3) are distributed over the territory, although more evenly, and an increase in concentrations was also noted near industrial sources (No. 2, 4) in the cities of Ulan-Ude (median = 3.8 mg/L; maximum = 11.5 mg/L) and Angarsk (median = 1.2 mg/L; maximum = 6.9 mg/L). A comparison of our results with the results obtained in [68–70] showed that the concentrations of nitrates in the snow cover of Ulan-Ude are 1.5 times higher than in Moscow (1.5 mg/L) and Izhevsk (2.4 mg/L) and close to the concentration in Vladivostok (3.4 mg/L). By analogy with sulfates, elevated concentrations of ni-

trates are recorded along the direction of the main transport (No. 8) up to 23 mg/L (median = 2.4 mg/L), and on the ice of the Southern Basin of Lake Baikal (No. 12), up to 9.1 mg/L (median = 2.5 mg/L).

If we consider the marker substances of other industrial enterprises, we can see a similar picture (Figure 3). For example, sodium and fluorine, markers of aluminum smelting [72,73], are concentrated near the aluminum plant of the city of Shelekhov. The maximum concentrations are 29.3 mg/L (Na) and 50.7 mg/L (F). The median fluoride content in Shelekhov's snow cover (3.8 mg/L) is close to the values in Bratsk (from 3 to 6 mg/L) [74] and Krasnoyarsk [75], where large aluminum plants are also located. When moving away from cities, there is a sharp decrease in fluorine concentrations to 0.03–0.2 mg/L, and sodium to 0.1–0.2 mg/L.

The maximum of chlorine ions is localized near the Angarsk Polymer Plant (Figure 3). This is due to the fact that chlorine is used in the production of polyvinyl chloride, the main product of this enterprise. Another significant source of chlorine in the region is anti-icing reagents for roadway treatment [76,77]. This determines that the highest concentrations of chlorine ions are localized in cities (sites No. 1–5). The median chlorine content in the snow cover of Irkutsk is 1 mg/L, in Angarsk—0.8 mg/L, in other cities—0.4 mg/L. A slight increase in chlorine concentrations was determined along highways (sections—six, seven, nine and ten)—0.2 mg/L. Comparative analysis showed that chlorine concentrations in the studied area are significantly lower than in Moscow [68] (8.6 mg/L) and Vladivostok [70] (5.2 mg/L), and are comparable with the results obtained in the Komi Republic [71].

The maximum concentrations of calcium ions and the alkaline nature of the pH value of snow water were found near the cement plants in the city of Angarsk and the village of Kamensk, as well as near the Pereval Quarry, which develops the Slyudyanka marble deposit and supplies raw materials for cement production (Irkutsk, Russia).

### 3.2. Spatial Distribution of Atmospheric Impurities in the Snow Cover of the Baikal Basin

In order to understand the overall picture of soluble pollutants in the snow water of the Baikal Basin, the distribution of electrical conductivity was mapped (Figure 4). The map allowed estimating the acreage and pathways of transfer of emissions from industrial enterprises, as well as zoning the region according to the degree of soluble pollutants in water (aerosol load). As a result, increased levels of snow cover pollution were identified in cities as sources of pollution (Figure 4). The city of Angarsk, being the most prone to anthropogenic pollution, had an industrial zone with a length of more than 30 km and a total atmospheric emission of more than 110 thousand tons per year [78]. The atmospheric emissions from the Irkutsk agglomeration and the cities of Buryatia extended for tens of kilometers along the prevailing transfer pathway as well as along the main traffic flows and were locally determined near small settlements located on the coast. The plumes from industrial enterprises of the Irkutsk agglomeration were found on the ice of South Baikal in the area of Cape Kadilny (marked with an asterisk in Figure 4). The impact of emissions from the cities of Selenginsk and Kamensk, which are the sources of pollution in Buryatia, was visualized in the delta of the Selenga River (Figure 4).

For determining the level of pollution by acidic components of the snow cover, a schematic map of the distribution of the pH value of the snow waters of the Baikal Basin was constructed (Figure 5). It is known that the pH value of atmospheric precipitation is mainly due to the presence of nitrogen, carbon, and sulfur oxides in the atmosphere. The pH value of atmospheric precipitation in the background regions of the world is 5.1–5.5 [79]. In the cities of the region, where alkaline aerosol particles (ash) from coal-fired thermal power plants are emitted into the atmosphere, the pH value varied from 6.25 to 7.75. High pH values varying from 11.05 to 11.07 units were determined in snow water near cement production enterprises (Angarsk, Kamensk). Acid atmospheric fallout (pH~4–5) was identified in the snow water of Southern Baikal and in the area of the Baikal State Biosphere Reserve (4.7). The lowest values (4.44) were recorded on the ice of Southern Baikal (Kadilny-Mishikha cape), which could be associated with the transfer of sulfur and

nitrogen oxides from large thermal power plants located in the valley of the Angara River. The rest of Baikal (districts 12–14) had pH values varying from 5.25 to 6.5 (Figure 5).

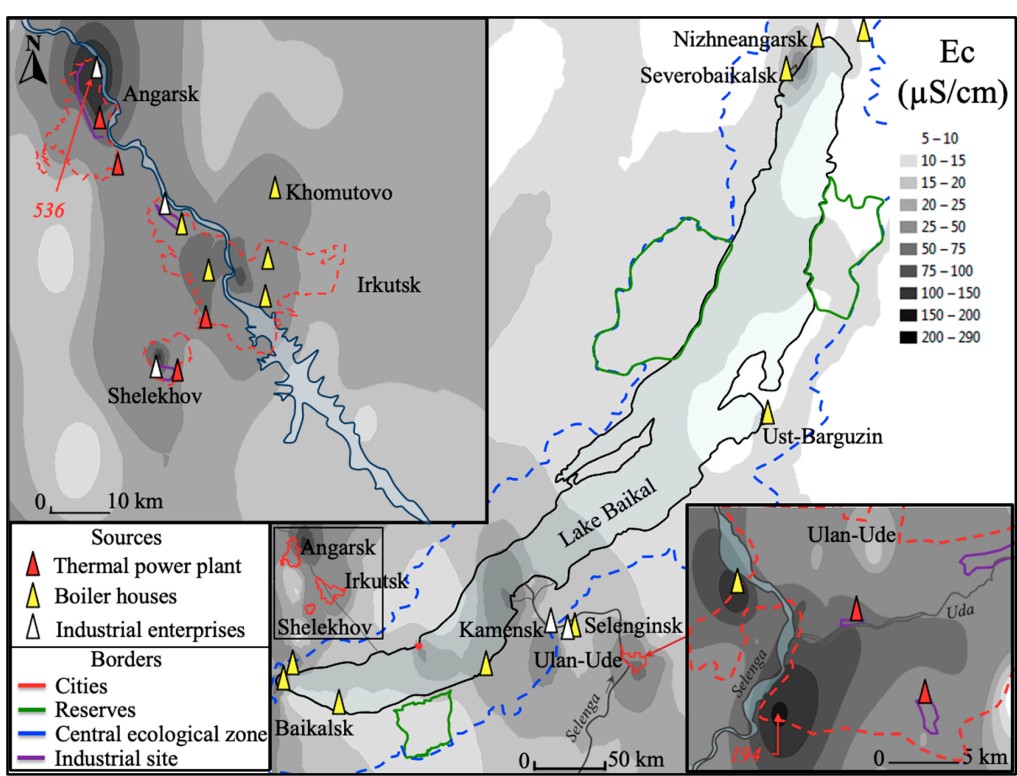

**Figure 4.** Schematic map of the spatial distribution of electrical conductivity in the snow water of Lake Baikal.

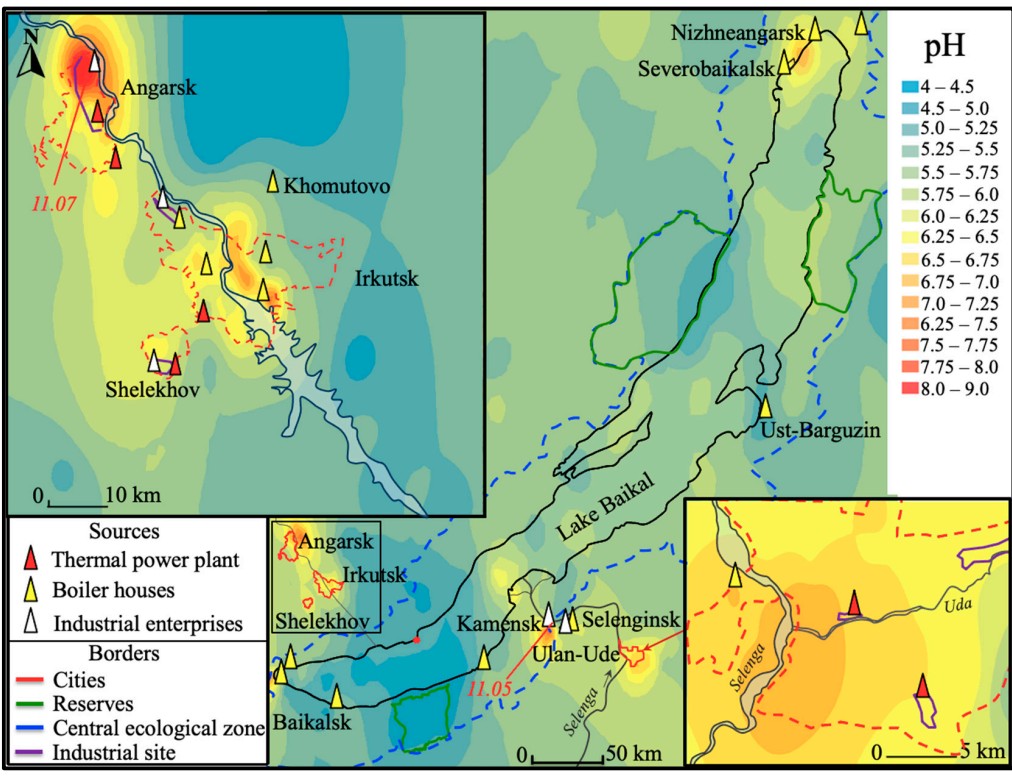

**Figure 5.** Schematic map of the spatial distribution of pH in the snow water of the Baikal basin.

Geovisualization of the results and zoning by environmental pressures is presented in Figures 6–8.

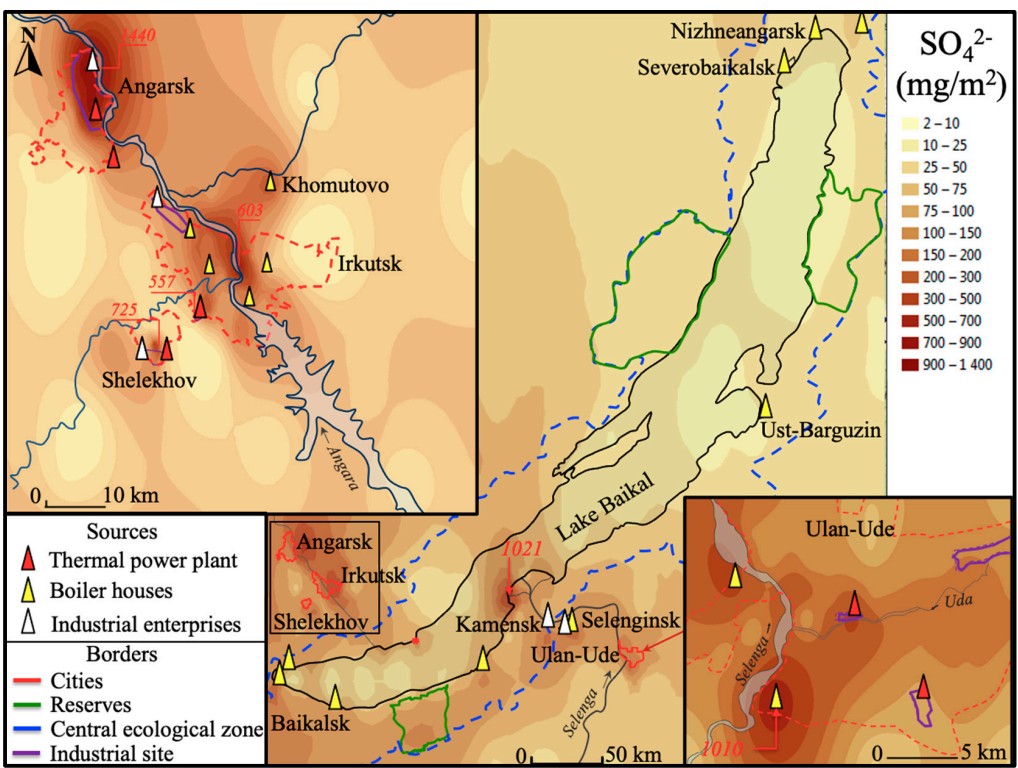

**Figure 6.** Schematic map of the spatial accumulation of sulfate ion ($SO_4^{2-}$) in the snow cover of the Baikal basin.

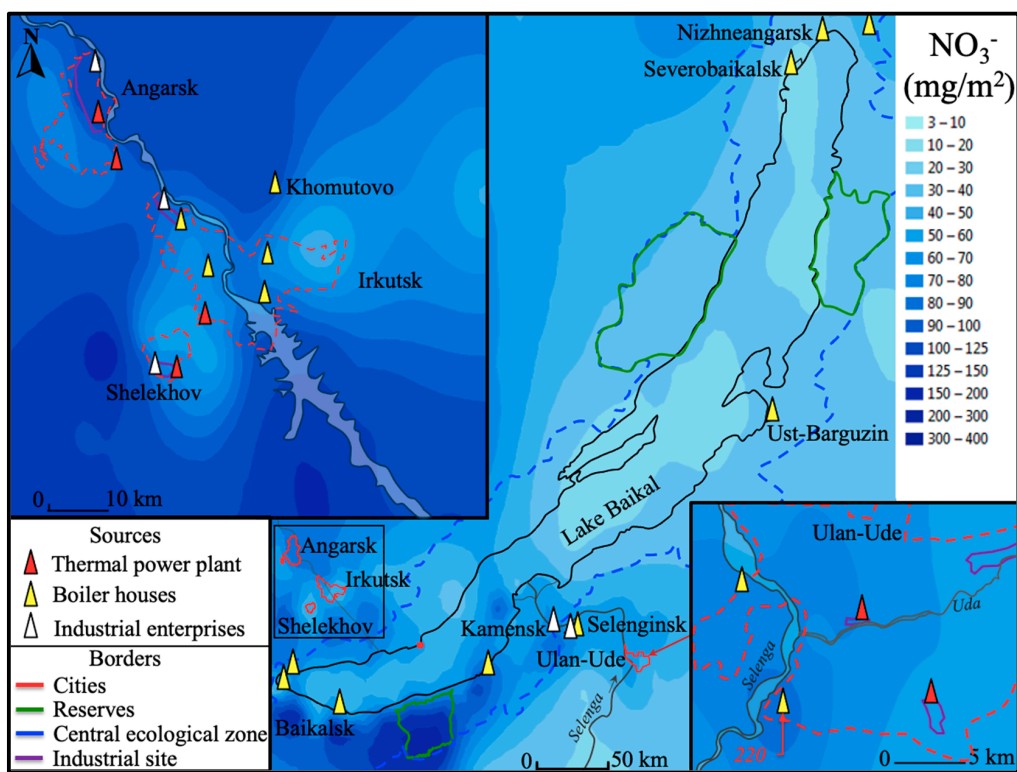

**Figure 7.** Schematic map of the spatial accumulation of nitrate ion ($NO_3^{-}$) in the snow cover of the Baikal basin.

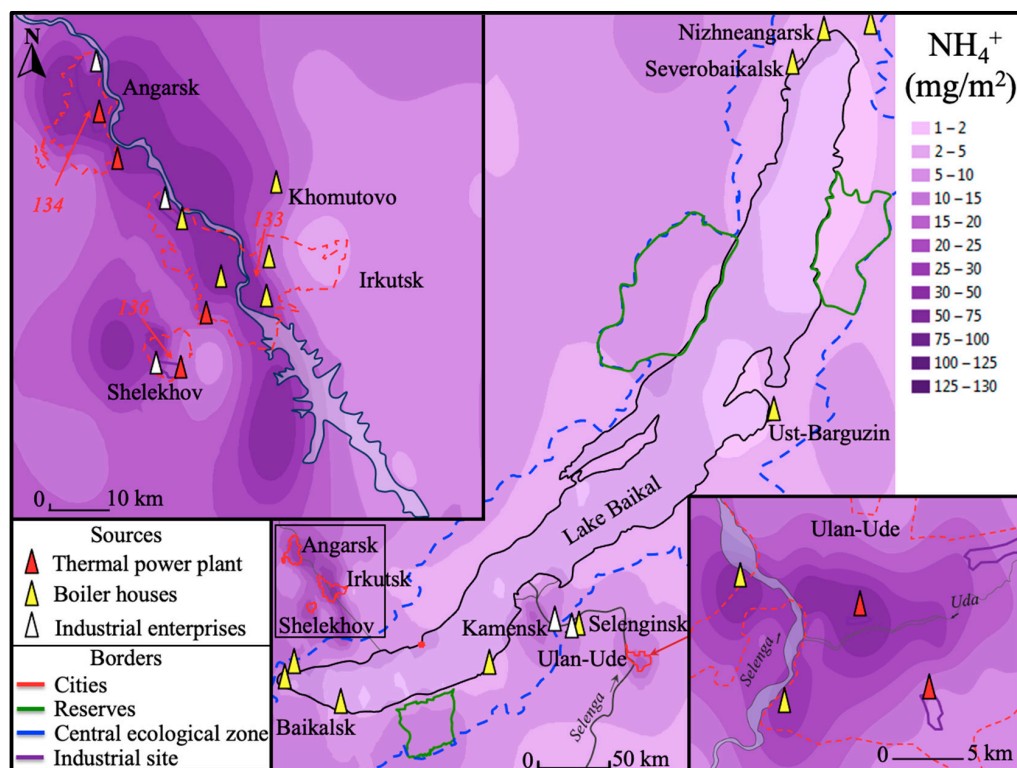

**Figure 8.** Schematic map of the spatial accumulation of ammonium ions ($NH_4^+$) in the snow cover of the Baikal basin.

After analyzing Table 1 and Figure 3, the main tracer substances for each group of sources were selected. The visual distribution of tracers in the snow cover of the Baikal basin allowed constructing fallout fields and determining the areas of pollutant distribution.

After a detailed examination of the Irkutsk agglomeration, which included three large industrial centers (Figure 1, districts 1–3) and the Irkutsk-rural region connecting them, it was found that high levels of tracer accumulation were visualized near the sources of pollution (Figures 6–8). A more detailed analysis of the spatial distribution of sulfates (Figure 6) allowed drawing a number of conclusions. The maximum accumulations of sulfates were located near heat sources, which was considered to be typical for both urban centers and rural ones (Figure 6). The highest level of sulfate accumulation was determined in the city of Angarsk and was localized near a large thermal power plant (from 1250 to 1440 mg/m$^2$). In Irkutsk, the maximum accumulations of pollutants were identified not only near the largest thermal power plant, located in the southwest of the city (550 mg/m$^2$), but also near small communal boiler houses (600 mg/m$^2$), which also largely participated in the city's heat supply. Local increases in sulfates were also identified near the village of Khomutovo (440 mg/m$^2$) due to a coal-fired boiler house located in it. A similar distribution pattern was observed on the territory of the Republic of Buryatia (Figure 1, district 4). The highest levels of tracer accumulation were noted in Ulan-Ude near large boiler houses located in the southwest (1010 mg/m$^2$) and west of the city and a thermal power plant located in the center (425 mg/m$^2$).

Another high accumulation of sulfate ions (1021 mg/m$^2$) was determined at a sufficient distance from the sources of pollution in the delta of the Selenga River (Figure 1, district 8) and was caused by the anthropogenic emissions from the Selenginsky Pulp and Cardboard Mill located at a distance of ~45 km to the southeast (Figure 6) from the delta and large settlements Kamensk and Kabansk (30–35 km from the delta).

On the territory of the Baikal State Natural Biosphere Reserve, located on the eastern coast of Southern Baikal at a distance of more than 100 km from large anthropogenic sources, an increase in the accumulation of sulfates in the snow cover was also identified

(200 mg/m$^2$). At the same time, according to Figure 3, their concentrations in the snow of the reserve were low, which was due to the large amount of moisture content exceeding the moisture content in cities by more than three times (Table 1).

Figure 7 demonstrates the spatial distribution of nitrates. The high-temperature combustion of fossil fuels by vehicles and industry is the main anthropogenic source of nitrates. Nitrates, unlike other considered compounds, were more evenly distributed over the studied area (Figure 7). An increased accumulation of nitrates was traced along the main pathway of transfer of air masses from stationary sources and along highways. The maximum accumulation of nitrates was recorded in the snow cover of the Baikal State Natural Biosphere Reserve (360 mg/m$^2$), which was due to the large moisture content in this area.

Figure 8 shows a schematic map of the accumulation of ammonium in the snow cover. The main sources of atmospheric ammonia were soil, agriculture, and, to a lesser extent, industry [80]. Sulfates and nitrates, as well as ammonium ions, were also tracers of long-range gas-phase transfer of impurities [81,82]. The distribution of ammonium was quite homogeneous in the central ecological zone of the lake, and local accumulation values were determined in industrial centers (Figure 1, districts 1–5).

According to the results of a geostatistical analysis of a large amount of data, the factors of remoteness from industrial enterprises, population density, and snow cover height were the most important factors affecting the amount of tracer accumulation in the snow cover of the Baikal region.

## 4. Conclusions

The research was devoted to the approbation of the method of geostatistical processing of a large amount of data for the construction of maps of the spatial distribution of atmospheric impurities in the snow cover of the Baikal basin.

Based on statistical processing of data and direct sampling of snow cover in large industrial centers and the Baikal basin, an inventory of sources affecting the composition of atmospheric precipitation in the region was performed. For each of the fourteen studied areas, a set of predominant tracer pollutants was obtained, and the levels of snow cover pollution varied greatly across the selected areas. The average level of snow pollution in the industrial centers of the region was two times higher than in the remote areas. Atmospheric impurities were transferred from the large regional cities (Irkutsk, Angarsk, and Shelekhov) tens of kilometers away from them along the prevailing wind directions. Snow pollution in the lake basin and on its coast was noted in settlements and along the valleys of the Angara and Selenga Rivers. The greater the distance from the pollution sources, the lower the concentrations and accumulated amounts of all substances analyzed in snow water.

Quantitative estimates of the input of soluble substances in the snow cover allowed for the zoning of the studied area according to the degree of anthropogenic load. Based on the analyzed indicators, the territory of the coast and water area of Southern Baikal was determined to be the most susceptible to anthropogenic pollution due to the transfer of atmospheric emissions from the industrial complexes of the Baikal region. Increased sulfate accumulations in the snow cover were recorded near coal heating sources. The accumulations of mineral forms of nitrogen in the snow cover in the studied area were more even and were associated with both heat and power sources and vehicles. A high level of accumulation of soluble substances in the snow cover was determined not only in areas with high concentrations of pollutants but also in areas with a high snow cover (the area of the Baikal Reserve with a snow cover depth of 80 to 105 cm).

The use of the geostatistical analysis method has proven to be an effective tool for conducting environmental impact assessments. In spatial terms, the methods used allowed for visually determining the range and intensity of the impact of a particular group of sources in the Baikal basin. The performed geostatistical analysis allowed visualization of stable pollution fields from enterprises producing cement, aluminum, etc. The localization of tracer substances in the vicinity of the source of pollution and in the direction of the main

transfer pathways of atmospheric emissions was determined both for single-industry towns (Selenginsk, Slyudyanka, Kamensk, and Shelekhov) and for the multiprofile industrial zone of the city of Angarsk. The revealed patterns of the spatial distribution of soluble pollutants in the snow cover, both for the background and urbanized areas of the studied region, will later be used while assessing the dispersion and zoning of the territory for other pollutants (heavy metals, persistent organic pollutants, etc.). With data on the volume of emissions and the amount of concentrations in the snow cover in the studied area, it is possible to calculate the deposition rates of various tracers on the underlying surface.

**Author Contributions:** Conceptualization, methodology, data analysis, writing, review and editing, Y.V.M.; data analysis, digital mapping, M.Y.S.; chemical analyses, O.G.N.; project administration, review and editing, T.V.K. All authors have read and agreed to the published version of the manuscript.

**Funding:** The research was performed on the topic of the state task of LIN SB RAS No. 0279-2021-0014 "Investigation of the role of atmospheric fallout on water and terrestrial ecosystems of the Baikal Basin, identification of sources of atmospheric pollution".

**Institutional Review Board Statement:** Not applicable.

**Informed Consent Statement:** Not applicable.

**Data Availability Statement:** Data used are available on request from the corresponding author.

**Acknowledgments:** The authors of this research would like to express their gratitude to the staff of the Baikal State Biosphere Reserve and the Limnological Institute (LIN SB RAS) for providing their assistance in sampling the snow cover. The authors are especially grateful to the staff of the Laboratory of Hydrochemistry and Atmospheric Chemistry of the Limnological Institute for performing part of the chemical analysis of snow water samples.

**Conflicts of Interest:** The authors declare no conflict of interest.

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
