# Peer review of "Ecological Zoning of the Baikal Basin Based on the Results of Chemical Analysis of the Composition of Atmospheric Precipitation Accumulated in the Snow Cover"

_applsci, doi:10.3390/app13148171_

Round 1

Reviewer 1 Report

Dear Authors

Thank you for submitting your manuscript entitled "Ecological zoning of the Baikal basin based on the results of chemical analysis of the composition of atmospheric precipitation accumulated in the snow cover" to Applied Sciences Journal. I have carefully reviewed your work and would like to provide you with some feedback.

Firstly, I appreciate the importance of your research topic and the effort you have invested in conducting the study. The analysis of the composition of atmospheric precipitation in the Baikal basin is a significant contribution to the field of ecology.

However, I regret to inform you that I am unable to recommend the publication of your manuscript in its present form. During my review, I identified several areas that require improvement to enhance the clarity and overall quality of your work.

One crucial aspect that needs attention is the presentation of results. While the figures included in the manuscript are informative, it is challenging to understand the specific details when they are only briefly mentioned in the text. I would suggest incorporating more numerical data and specific findings directly into the main text to provide a clearer understanding of your results.

Additionally, the discussion section of your article requires further development. A comprehensive discussion is essential for interpreting the results, drawing meaningful conclusions, and relating your findings to existing literature. I encourage you to expand on the implications of your research, discuss any limitations or uncertainties, and provide a more thorough analysis of the results.

I understand that these comments may require additional effort on your part. However, addressing these concerns will significantly improve the quality and impact of your manuscript. I believe that with the necessary revisions, your work has the potential to make a valuable contribution to the scientific community.

I hope you find my feedback constructive and helpful for the revision process. If you have any questions or would like further clarification on any specific points, please do not hesitate to reach out. I look forward to seeing an improved version of your manuscript in the future.

Thank you once again for considering Applied Sciences Journal for the publication of your research. I appreciate your dedication to advancing scientific knowledge in the field of ecology.

Best regards, 

NA

Author Response

Dear reviewer answers to your comments in the attachment

Reviewer 2 Report

The authors examined a method of geostatistical processing of large data for building maps showing the spatial distribution of atmospheric impurities in the Baikal basin’s snow cover. For this, the authors took more than 800 samples and considered open-access data and snow samples for a period of 6 years (March 2017-2022). These were studied using GIS methods. The authors reported two times higher snow pollution in the industrial centers of the region relative to the remote region. They concluded that the southern Baikal which is industrial is the most susceptible to anthropogenic pollution.

I commend the authors’ effort in getting these data over 6 years. However, the description of the geographic information system (GIS) is unclear and the rationale for its use in this study was not established in the methodology. The authors only mentioned ‘GIS’ methods in the conclusion as the best methods but what are GIS methods? Why were other methods of geospatial technology e.g. remote sensing not considered in this study?

Author Response

Dear reviewer, thank you very much for your time spent on evaluating our work. Answers to your questions in the attachment

Round 2

Reviewer 1 Report

Dear Authors,

Thank you for your response and addressing the concerns raised regarding the absence of a discussion section in your paper. However, it is still important to include a proper discussion section in order to enhance the scientific rigor and completeness of your research.

The discussion section plays a crucial role in interpreting and contextualizing the research findings, comparing them with existing literature, discussing limitations and potential implications, and proposing further avenues for research. Without a discussion section, the paper lacks a comprehensive analysis and synthesis of the results.

You mentioned that large-scale studies like yours have not been conducted in your region, making it inappropriate to solely compare your findings with those from industrial centers in other regions. While this is a valid point, it is still essential to situate your findings within the broader scientific context and discuss their implications. It would be valuable to discuss how your findings contribute to or differ from the existing body of knowledge, even if there are no direct comparisons available.

Additionally, the limited bibliography provided in the results (just 3) section does not fulfill the requirements of a comprehensive discussion. We encourage you to include a broader range of references that are relevant to your research topic. This will help support your discussion and demonstrate a thorough understanding of the existing literature.

I kindly request that you revise your paper to include a dedicated discussion section that critically analyzes and interprets your findings, addresses limitations, discusses implications, and suggests potential avenues for further research. Furthermore, please ensure that your bibliography is expanded to incorporate a broader range of references that are pertinent to your study.

Thank you for your attention to these important revisions. 

NA

Author Response

Good afternoon, dear Reviewer! Thank you so much for your edits. We tried to take into account your comments and redesigned the "Discussions" section to make it more detailed.  

Please see the corrected version of the manuscript. We really hope that we were able to fulfill your wishes.  We are ready for further dialogue. 

 P.S. The last edits are highlighted in green.

Reviewer 2 Report

None

None

Author Response

Dear reviewer, thank you  so much!much for the work you have done and the attention you have paid to our article.

Round 3

Reviewer 1 Report

Many thanks for your revision. I have no further comments.

NA

Author Response

Dear reviewer, thank you  so much! Much for the work you have done and the attention you have paid to our article.
